# Pan-cancer segmentation in CT scans based on dynamic convolution with nnSAM

Ruikun Gao[0009−0005−8285−1267], Xinru Li[0009−0002−4509−2407], Jingyan Wang[0009−0009−0264−4524], Shaohua Zheng[✉]

College of Physics and Information Engineering, Fuzhou University, Fuzhou 350108, Fujian, China, sunphen@fzu.edu.cn

**Abstract.** Accurate and efficient segmentation of tumor locations from medical images is essential for clinical applications such as disease diagnosis and treatment planning. In this paper, we propose a method for whole-body pan-cancer segmentation based on **nnSAM** architecture combined with dynamic convolution. Our approach integrates the powerful feature extraction capability of SAM model, the powerful auto-configuration design capability of **nnUNet**, and the dynamic convolution to improve the representation capability of the model. In addition, in order to improve the accuracy of segmentation, we introduce attention mechanism in **nnSAM** architecture. This mechanism allows the network to focus on highlighted areas and suppress irrelevant background areas, thereby improving overall segmentation performance. We evaluate our proposed approach on the MICCAI FLARE 2024 Testing dataset, achieving a mean DSC of 16.34 % and a mean NSD of 11.66 %.

**Keywords:** Pan-cancer segmentation · SAM · dynamic convolution.

## 1 Introduction

Medical image segmentation is important for clinical applications, including disease diagnosis, treatment planning, and image-guided interventions. Accurate and efficient segmentation of whole-body cancers from medical images is important for assessing organ function, detecting abnormalities, and guiding surgical procedures. However, whole-body pan-cancer segmentation is a challenging task due to the uncertainty of cancer location, tumor size, noise and artifacts. In addition, labeled data is difficult to obtain, and unlabeled data is easier to access. In recent years, deep learning-based methods have been widely used in pan-cancer segmentation and achieved good results, among which nnUNet [11] is one of the most commonly used methods. However, nnUNet's high resource consumption and low inference speed cannot meet Challenge requirement of fast and low resource. In this work, the main contributions are summarized as follows:

(1)We use the nnSAM [14] segmentation framework, which can effectively integrate the powerful feature extraction capability of SAM model and the powerful automatic configuration design capability of nnUNet.

(2)Embedding dynamic convolution in the nnSAM framework improves model representation without increasing computational complexity.

(3)An attention mechanism is embedded in the nnSAM architecture to better capture significant areas and suppress irrelevant background areas.

## 2   Method

### 2.1   Preprocessing

Integrating nnUNet into the nnSAM allows automated network architecture and hyperparameter configuration, making it highly adaptable to the unique and specific features of each medical imaging dataset. This adaptive capability starts from a self-configuration process that automatically adjusts the nnUNet encoder's architecture to suit training dataset characteristics including the dimensions of the medical images, the number of channels, and the number of classes involved in the segmentation task.

### 2.2   Proposed Method

As shown in Fig.1, Our method follows the standard nnSAM design to achieve pan-cancers segmentation. Specifically, we introduce attention mechanisms into segmentation networks to enhance their ability to focus on regions of interest while suppressing unrelated background regions. The attention module can be well embedded in skip connections, which can improve the performance of the model without adding too much computation.

**Network Architecture Details.** Light-weight convolutional neural networks (CNNs) suffer performance degradation as their low computational budgets constrain both the depth (number of convolution layers) and the width (number of channels) of CNNs, resulting in limited representation capability. As shown in Fig.2, Dynamic Convolution [3], a new design that increases model complexity without increasing the network depth or width. Instead of using a single convolution kernel per layer, dynamic convolution aggregates multiple parallel convolution kernels dynamically based upon their attentions, which are input dependent. Assembling multiple kernels is not only computationally efficient due to the small kernel size, but also has more representation power since these kernels are aggregated in a non-linear way via attention. In addition, We introduce the CBAM [25] attention mechanism into the segmentation network to enhance its ability to focus on region of interest while suppressing irrelevant background region.

**Loss function.** we use the summation between Dice loss and cross-entropy loss because compound loss functions have proven robust in various medical image segmentation tasks [15].

**Strategies to Deal with the Partial Labels.** The dataset provided by the FLARE 2024 challenge included whole-body cancer segmentation CT scans with partial labels, and we did not know which cancer was labeled in each case,

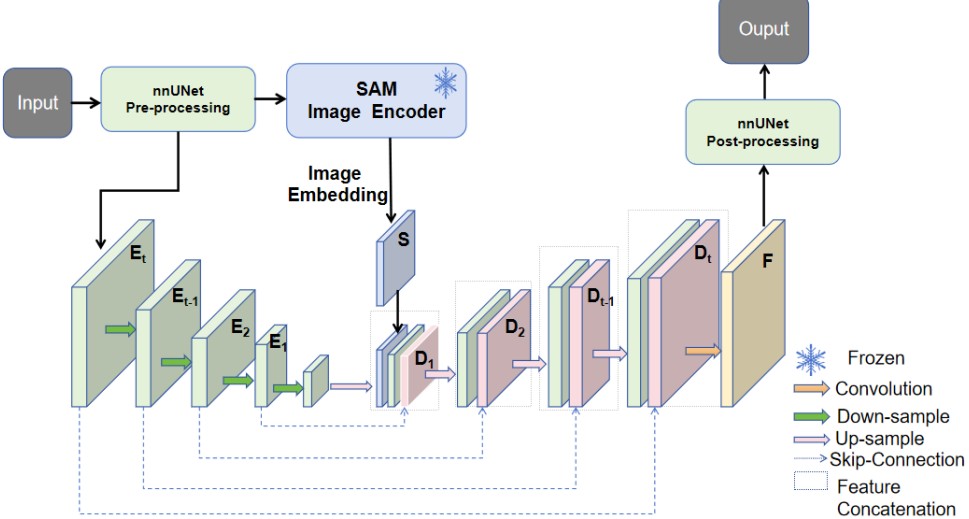

**Fig. 1.** Network architecture: Embedding SAM into nnUNet

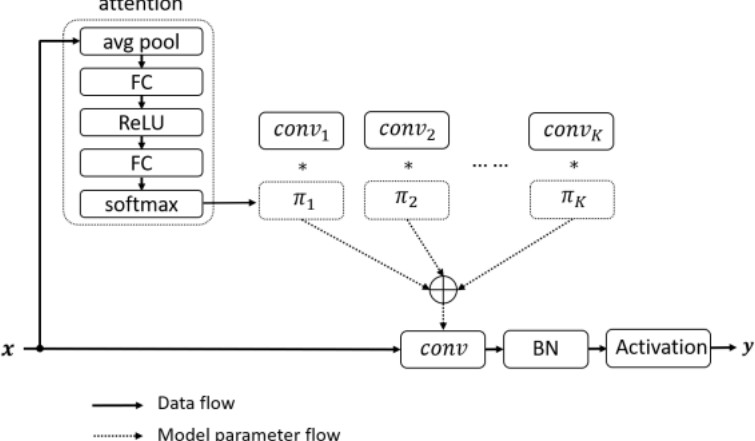

**Fig. 2.** Network module

but due to the amount of the data was sufficient, our method did not make special treatment for the data with partial labels.

**Strategies to use the unlabeled images.** Unlabeled images were not used.

### 2.3   Post-processing

We use connected component-based post-process to remove noise and isolated pixels and improve segmentation results.

## 3   Experiments

### 3.1   Dataset and evaluation measures

The segmentation targets cover various lesions. The training dataset is curated from more than 50 medical centers under the license permission, including TCIA [4], LiTS [2], MSD [23], KiTS [8,10,9], autoPET [7,6], TotalSegmentator [24], and AbdomenCT-1K [20], FLARE 2023 [19], DeepLesion [27], COVID-19-CT-Seg-Benchmark [18], COVID-19-20 [22], CHOS [13], LNDB [21], and LIDC [1]. The training set includes 4000 abdomen CT scans where 2200 CT scans with partial labels and 1800 CT scans without labels. The validation and testing sets include 100 and 400 CT scans, respectively, which cover various abdominal cancer types, such as liver cancer, kidney cancer, pancreas cancer, colon cancer, gastric cancer, and so on. The lesion annotation process used ITK-SNAP [28], nnU-Net [12], MedSAM [16], and Slicer Plugins [5,17].

The evaluation metrics encompass two accuracy measures—Dice Similarity Coefficient (DSC) and Normalized Surface Dice (NSD)—alongside two efficiency measures—running time and area under the GPU memory-time curve. These metrics collectively contribute to the ranking computation. Furthermore, the running time and GPU memory consumption are considered within tolerances of 45 seconds and 4 GB, respectively.

### 3.2   Implementation details

**Environment settings** The development environments and requirements are presented in Table 1.

**Training protocols.** The Training protocols and details (e.g., batch size, epoch, optimizer) are presented in Table 2. In the training process, the batch size is 4 and the patch size is fixed as $3\times192\times192$ for optimization, we train it for 1000 epochs using Adam with a learning rate of 0.001 and the learning rate reduction strategy using CosineAnnealingLR.

**Table 1.** Development environments and requirements.

| | |
|---|---|
| System | Windows10/Ubuntu 20.04.4 LTS |
| CPU | Intel(R) Core(TM) i9-12900K CPU@3.20 GHz |
| RAM | 4×4GB; 2.4MT/s |
| GPU (number and type) | One RTX 2080Ti 8G |
| CUDA version | 11.4 |
| Programming language | Python 3.9.18 |
| Deep learning framework | torch 2.1.1, torchvision 0.16.1 |
| Specific dependencies | pandas, scipy, collections |

**Table 2.** Training protocols.

| | |
|---|---|
| Network initialization | |
| Batch size | 4 |
| Patch size | $3 \times 192 \times 192$ |
| Total epochs | 1000 |
| Optimizer | Adam |
| Initial learning rate (lr) | 0.001 |
| Lr decay schedule | CosineAnnealingLR |
| Training time | 72 hours |
| Loss function | Dice plus CE |
| Number of model parameters | 74.2M[1] |
| Number of flops | 15.6G[2] |
| $CO_2$eq | 1 Kg[3] |

**Table 3.** Quantitative evaluation results.

| Methods | Public Validation | | Online Validation | | Testing | |
|---|---|---|---|---|---|---|
| | DSC(%) | NSD(%) | DSC(%) | NSD(%) | DSC(%) | NSD (%) |
| Algorithm1 | 19.87 ±31.54 , | 13.64 ± 23.78 | 18.76 ±27.64 , | 13.44 ± 22.78 | 16.34± 29.45 , | 11.66± 22.66 |

## 4    Results and discussion

As shown in Table 3., Pan-cancer segmentation is a very challenging task because of the uncertainty due to the large number of tumor types, wide distribution and inconsistent lesion size.

### 4.1    Quantitative results on validation set

Table 4. shows the runtime and resource consumption of our method. It clearly shows that our approach can basically meet the requirements in terms of run time, and as the number of layers increases, the cost of resources and run time increases.

**Table 4.** Quantitative evaluation of segmentation efficiency in terms of the running them and GPU memory consumption. Total GPU denotes the area under GPU Memory-Time curve. Evaluation GPU platform: NVIDIA RTX 2080Ti (8G) .

| Case ID | Image Size | Running Time (s) | Max GPU (MB) | Total GPU (MB) |
|---------|------------|------------------|--------------|----------------|
| 0001 | (512, 512, 55) | 15.64 | 1570 | 14158 |
| 0051 | (512, 512, 100) | 20.45 | 1570 | 21651 |
| 0017 | (512, 512, 150) | 28.79 | 1570 | 33231 |
| 0019 | (512, 512, 215) | 38.45 | 1570 | 46742 |
| 0099 | (512, 512, 334) | 51.84 | 1570 | 55541 |
| 0063 | (512, 512, 448) | 57.79 | 1570 | 62594 |
| 0048 | (512, 512, 499) | 61.51 | 1570 | 84912 |
| 0029 | (512, 512, 554) | 72.54 | 1570 | 95581 |

### 4.2    Qualitative results on validation set

Fig. 3 The segmentation results of our method are shown. This clearly shows that our method can obtain better segmentation results in large lesion segmentation compared to small lesion segmentation. However, there are still some considerations for the generalization and stability of the lesion types and cancer types with great differences.

### 4.3    Segmentation efficiency results on validation set

We evaluated the segmentation efficiency on validation set, some of the results are shows in Table 4.

### 4.4    Results on final testing set

As shown in Table 3., our method achieves a mean DSC of 16.34% and a mean NSD of 11.66% on the FLARE 2024 final testing set.

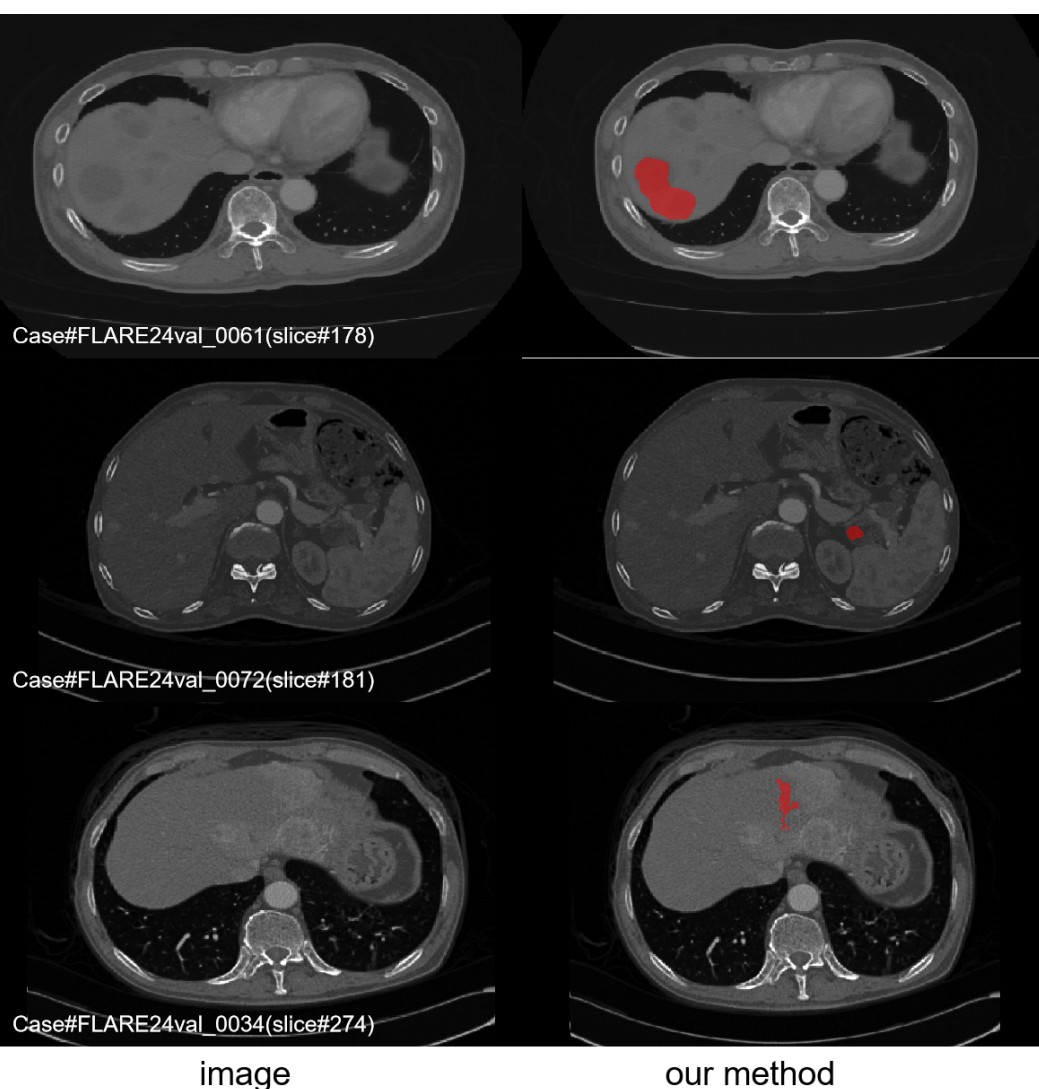

**Fig. 3.** Example cases from the MICCAI FLARE 2024 validation set. Our method does not achieve good segmentation results on the validation set, and here are just one examples that seem to have slightly better segmentation results (No. 0061) and two examples that have poor segmentation results (No. 0034 and No. 0072).

### 4.5   Limitation and future work

Our proposed pan-cancer segmentation method does not achieve good segmentation results, the limitations of our method are that we do not make full use of unlabeled data, and training on partially labeled data may introduce noise and inconsistencies during the training process, resulting in degraded model performance. In addition, pan-cancer segmentation covers a wide range of lesions, with significant differences between lesion types.

Therefore, in the future, we will focus on introducing techniques such as active learning or semi-supervised learning to iteratively select and annotate the most informative instances, thereby improving the performance of the model when partially labeling the data. It is more likely to train a good segmentation model based on a large medical image model, and use transfer learning technology to adjust our model, so as to achieve a time-saving effect.

## 5   Conclusion

In this work, based on the nnSAM framework, we integrate SAM's powerful feature extraction capability with nnunet's automatic configuration capability and combine dynamic convolution and attention mechanism to improve the expression ability of the model in pan-cancer segmentation. The challenge of incomplete data annotation is overcome by using partial labeled data in training. Future studies could further extend this approach and validate it in a wider range of medical image segmentation tasks.

**Acknowledgements** The authors of this paper declare that the segmentation method they implemented for participation in the FLARE 2024 challenge has not used any pre-trained models nor additional datasets other than those provided by the organizers. The proposed solution is fully automatic without any manual intervention. We thank all data owners for making the CT scans publicly available and CodaLab [26] for hosting the challenge platform.

This work was funded by the National Natural Science Foundation of China (62271149) Fuzhou Science Technology Project(2023-P-001), Fujian Science Technology Project(2022L3003, 2021H0013, 2020Y9091, 2022Y4014).

## Disclosure of Interests

The authors declare no competing interests.

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
