# OpenReview forum: "Pan-cancer segmentation in CT scans based on dynamic convolution with nnSAM"
_MICCAI.org/2024/Challenge/FLARE — Submitted to FLARE 2024_

### Official Review · Reviewer_Q4sc · 2025-03-02
**polish figures and tables**

**Rating:** 6
**Confidence:** 5

**Review:**

Abstract: “We evaluated our proposed approach on the MICCAI FLARE 2024 validation dataset with mean DSC and mean NSD.”
Please show the numbers here

Fig. 1. Add detailed description to the caption.

Table 1-2. Fill out the missed rows. If code is not available, please delete this row.

Fig 3. Adjust window width and level

---

> ### Author Response · Authors · 2025-03-29
>
> Thank you for your valuable advice
> We have adjusted our paper in response to your comments

---

### Official Review · Reviewer_Hqsy · 2025-03-02
**Typos and style**

**Rating:** 6
**Confidence:** 5

**Review:**

In the manuscript, some sentences lack a space after punctuation marks, for example:
"Ruikun Gao[0009−0005−8285−1267] ,Xinru Li"
"As shown in Figure.1,Our method follows..."
"...background regions.The attention module..."

Please use numbered lists instead of hyphens, such as
(1) We use the nnSAM...
(2) Embedding dynamic convolution...

Typos and style: There are many typos or formatting issues, such as
“As shown in Figure.1, … “ Figure.1 change to Fig. 1.
"As shown in Table.3," Table.3 change to Table 3.
"...better segmentation results (No. 0061 ) and..." (No. 0061 ) change to (No. 0061)

Please carefully read the whole paper and improve the writing quality.

---

> ### Author Response · Authors · 2025-03-29
>
> Thank you for your valuable comments.
>
> 1. The problem of leaving one space after adjusting the punctuation mark
>
> 2. The hyphen has been adjusted for this problem
>
> 3. Picture table spelling style error problem has been adjusted

---

### Official Review · Reviewer_s8xt · 2025-03-03
**Review of "Pan-cancer segmentation in CT scans based on dynamic convolution with nnSAM"**

**Rating:** 6
**Confidence:** 5

**Review:**

This research proposes a method for whole-body pan-cancer segmentation based on nnSAM architecture combined with dynamic convolution.
1. Please adjust all CT images to the proper window width and level (e.g., 400/40 for abdomen CT).
2. Please use "3×192×192" instead of "3*192*192".
3. There should be a space after punctuation, e.g., "presented in Table 2. In the " instead of "presented in Table 2.In the ".
4. There are some inconsistencies in the tenses in the abstract

---

> ### Author Response · Authors · 2025-03-29
>
> Thank you for your valuable comments.
>
> 1. Because task 1 is a whole-body segmentation task, all CT images are not uniformly adjusted to the same window width and window position.
>
> 2. The issue of multiplicators has been adjusted
>
> 3. The punctuation has been adjusted for this problem.
>
> 4. The tense problem has been adjusted.

---

### Decision · Program_Chairs · 2025-03-20

**Decision:**

Accept

**Comment:**

Please reduce the size of Fig 3 to make it fit to the template.